# Design and Validation of Lifetime Extension Low Latency MAC Protocol (LELLMAC) for Wireless Sensor Networks Using a Hybrid Algorithm

**Tao Hai [1,2]**, **Jincheng Zhou [1,3,4]**, **T. V. Padmavathy [5]**, **Abdul Quadir Md [5,*]**, **Dayang N. A. Jawawi [2]** and **Muammer Aksoy [6]**

1   School of Computer and Information, Qiannan Normal University for Nationalities, Duyun 558000, China
2   Faculty of Computing, Universiti Teknologi Malaysia (UTM), UTM Skudai, Johor Bahru 81310, Johor, Malaysia
3   Key Laboratory of Complex Systems and Intelligent Optimization of Guizhou Province, Duyun 558000, China
4   Key Laboratory of Complex Systems and Intelligent Optimization of Qiannan, Duyun 558000, China
5   School of Computer Science and Engineering, Vellore Institute of Technology, Chennai 600127, India
6   Computer Information Systems Department, Ahmed Bin Mohammed Military College, Doha P.O. Box 22988, Qatar
*   Correspondence: abdulquadir.md@vit.ac.in

**Abstract:** As the battery-operated power source of wireless sensor networks, energy consumption is a major concern. The medium-access protocol design solves the energy usage of sensor nodes while transmitting and receiving data, thereby improving the sensor network's lifetime. The suggested work employs a hybrid algorithm to improve the energy efficiency of sensor networks with nodes that are regularly placed. Every node in this protocol has three operating modes, which are sleep mode, receive mode, and send mode. Every node enters a periodic sleep state in order to conserve energy, and after waking up, it waits for communication. During the sleep mode, the nodes turn off their radios in order to reduce the amount of energy they consume while not in use. As an added feature, this article offers a channel access mechanism in which the sensors can send data based on the Logical Link Decision (LLD) algorithm and receive data based on the adaptive reception method. It is meant to select acceptable intermediary nodes in order to identify the path from the source to the destination and to minimize data transmission delays among the nodes in the network scenario. Aside from that, both simulation and analytical findings are used to examine the activity of the suggested MAC, and the created models are evaluated depending on their performance. With regard to energy consumption, latency, throughput, and power efficiency, the result demonstrates that the suggested MAC protocol outperforms the corresponding set of rules. The extensive simulation and analytical analysis showed that the energy consumption of the proposed LELLMAC protocol is reduced by 22% and 76.9% the end-to-end latency is 84.7% and 87.4% percent shorter, and the throughput is 60.3% and 70.5% higher than the existing techniques when the number of node is 10 and 100 respectively.

**Keywords:** energy efficiency; heterogeneous network; network topology; throughput; wireless sensor networks

## 1. Introduction

Wireless Sensor Networks (WSNs) can be characterized as a collection of interconnected sensor nodes [1], which seem to be miniature devices with constrained power and memory capabilities. There is a growing need to create more effective sensor networks because of applications from habitat monitoring, and logistics to animal monitoring. Sensor networks differ from other ad hoc networks in particular because of the characteristics of WSNs, such as restrictions on sources such as energy, processing speed, and storage that are available. In addition to these limitations, WSNs are also subject to a variety of constraints, such as the

variable node deployment density and potentially dangerous environmental conditions. Perhaps the most significant parameter for measuring the sensor network performance is the network lifespan. The usage of each finite resource must, of course, be taken into account in a resource-constrained context. Network lifetime, however, has the unique position of forming an arbitrary limit for the usability of the sensor nodes as a metric of energy usage.

Today's rapid adoption of Wireless Sensor Networks (WSNs) and incorporation of Internet of Things (IoT) technology have made it possible for their use to expand in a number of industrial fields.in the nation. Modifications in Medium Access Control (MAC) protocols, which are considered essential for WSNs-IoT, are just one of the many factors that influence the success of WSN development. Considerations for reducing the energy consumption, performance, and scalability for a big scale are just a few. Many protocols, though, only take a limited approach to handling medium access when addressing this issue. A modern analysis of newly proposed WSN MAC techniques is presented in this study. Various strategies and methods are suggested to improve the primary performance elements.

## 2. Related Works

The latest methods for energy management and prolonging the lifespan of sensor networks are discussed in this section [2,3]. In recent years, a slew of MAC protocols have been created. By the manner in which they regulate access to the media, currently- available MAC protocols may be categorized. Because of the narrow-band nature of communication in WSNs, current FDMA methods are not recommended due to the existence of these limitations. Because the channel bandwidth is restricted, FDMA methods do not provide great levels of efficiency in WSNs. CDMA and OFDMA methods, on the other hand, have proved to be effective and are now being utilized in cellular networks, despite the fact that they are not usually favored owing to their low-cost efficiency.

While these systems need more complexity and energy usage than conventional WSNs, they have a distinct advantage over them. The end-to-end response time, data transmission and reception dependability, and drop rate are all important considerations for industrial and technical applications [4–6]. A protocol that uses the TDMA method for reliable multicast data processing on both the transmitter and the receiver side is described below. This protocol is particularly developed for wireless sensor networks (WSNs) and is supported on contention and energy efficiency [5,6]. This protocol takes advantage of the sleep state of wireless radios in order to trade of the energy and to increase throughput and latency.

Due to the fact in S-MAC protocol have fixed listening time, even if there is only a small quantity of traffic, it results in energy waste. According to [7], the Dynamic MAC protocol may be used to reduce the interference between WSNs and WLANs. This technique is used to anticipate the optimal communication route for establishing a coexistence between the wireless nodes that are operating in the ISM spectrum band. If, on the other hand, there is a great deal of traffic, a constant length may not result in a significant amount of traffic. Consequently, the Timeout MAC (T-MAC) protocol [8] was developed. Since it utilizes an active period that adjusts to the traffic density, the TMAC protocol differs from the S-MAC protocol. CSMA with a preamble sampling method is a variation of B-MAC [9], which is a form of CSMA. As a result, B-MAC is extremely customizable and may be implemented with a minimal code and memory. Also, an example of a protocol that takes use of the duty cycle to save energy is the Routing-Enhanced MAC (RMAC) protocol [10]. Comparing this protocol to the S-MAC standard, it seeks to reduce end-to-end latency and prevent congestion.

The number one intention of RMAC is to fit the sleep/wake periods of the nodes with the course of the sensor statistics to make certain ensure that the packet is sent to its destination in an unused operational cycle. RMAC searches for better lantecies, which are often visible in MAC protocols that utilize duty cycles, to enhance performance. The receiver-initiated MAC (RI-MAC) protocol is a contention-based protocol, where the transmission is always initiated with the aid of the individual that is receiving the records [11,12] and the transmission is constantly a success. Because of a mathematical version, a network performance for the SMAC protocol was developed, and the take a look at has usually centered on topology.

It is the receiver's duty to control its functioning, which includes whilst to just accept information and identify collisions in addition to when to and re-cover the facts misplaced in transmission. TDM has several advantages over other protocols, including Medium Access Control (LMAC) protocol [13] and the cell Low Weight Medium get entry to manipulate (ML-MAC) protocol [14]. In different phrases, these protocols provide collision-free conversation while additionally managing power effectively. In addition, they are able to establish transmittal schedules in a dispersed way. However, in both protocols, the slot size is constant, and its allocations are likewise are consistent, ensuing in inefficiency in terms of bandwidth use. In line with [15], the DSME MAC protocol become advanced to prevent the channel from being overloaded because of the transport of acknowledgement packets without any extra CCA. In this paper, the IEEE 802.15.4 single channel operating mode is extended to multichannel the operation for statistics transmissions, as a consequence decreasing the effects of mutual interference because of the heterogeneous nodes inside the network. The authors in [16] Has cautioned an Analytical approach to study the queue-size brief distribution, a mathematical approach is used. A better site visitors magnificence Prioritization based totally provider feel a couple of access/Collision Avoidance (TCP-CSMA/CA) technique for precedence channel get admission to in heterogeneous networks became counseled by using the authors [17]. However, the verbal exchange is receiver-pushed, as opposed to transmission-driven, as in TDMA [18], so therefore strength utilization is kept to an absolute minimum.

Based on the Euclidean and distance between each sensing node and the cluster centroid, the K-mean clustering assigned the sensing nodes to particular clusters. This type of clustering was selected for this investigation because it enables the CH and nodes to be located to one another, minimizing the energy usage [19].

In addition to the current one, it makes use of uses several channels to increase the achieved throughput that can be achieved also reducing the delivery delay that may be experienced. However, the main cons downside of the Y-MAC method is that it suffers from the same flexibleness and measurability problems as TDMA, and that it needs sensor nodes to have several radio channels for its sensor nodes to function properly.

Since the layered architecture offers shared medium access while all other upper layers, networks, and transport are constrained to the MAC layer to meet secure QoS requirements, the Medium Access Control (MAC) layer's improvement took up a significant portion of them [20]. In order to guarantee QoS parameters in WSN networks, a proposed MAC layer upgrade is examined in this research.

The MAC layer, which controls the shared medium used by WSNs to transport data, employs a Carrier Sense Multiple Access/Collision Detection algorithm to reduce data loss by preventing collisions. The majority of the energy used in WSN networks is used for transmission [21]. As a result, the MAC layer is regarded as the core component of WSN application framework. Important QoS factors such as delays, bandwidth, the rate of packet delivery (PDR), and power consumption, largely depend on the design of the MAC layer.

The importance of MAC in networking grows as the number of devices increases since it is essential for organizing a device's accessibility during the allotment to the medium and avoiding collision [22]. There are two primary MAC protocol types: TDMA (Time Division Multiple Access) and CSMA (Carrier Sense Multiple Access). Due to the fact that TMDA primarily relies on a time-triggered paradigm, it requires extremely accurate duty cycling and requires the coordinator to periodically broadcast a beacon packet even when the detectors have no data to send.

The QoS parameter performance and the available bandwidth are prioritized in the MAC protocol design priority in the design of the MAC protocol, which is mostly dependent on the application, such as wired network computer networks [23]. In battery-powered WSNs, the effectiveness of energy usage and the lifetime of the network are key challenges. Load balance and power neutrality, nevertheless, are seen as two of the biggest problems in energy harvesting WSNs [24].

The focus of the literature has been on MAC protocol improvement in wireless sensor networks for mission-critical uses in industries like health care and disaster alerting

and monitoring [25,26]. General techniques, game hypothesis techniques, heuristic-based methods, meta-heuristic-based perspectives, machine learning-based methodologies, and MAC schedules in cross-layer approaches are some of these subcategories.

Energy management, lifespan extension, and end-to-end delay are all factors that influence the quality of service (QoS) of an EH-WSN via the MAC protocol. The residual power of the node is considered in this protocol for the purpose of prolonging the network's life. For reducing the time it takes for data to travel from point A to point B due to multi-hop routing, pipeline-forwarding media access control (MAC) protocols such as Routing-enhanced duty-cycling media access control (R-MAC), PRIMAC, and Reduced Pipelined Media Access Control (RP-MAC), and Routing enhanced duty-cycle media access control (R-MAC) was proposed.

According to [27], the energy-efficient MAC protocol was employed by allowing a scheduling table for the node's sleep/wake-up period and segmenting the channel into TDMA slots with CSMA/CA support for each slot. This protocol experiences overhearing because it has to refresh the scheduling table, much like the S-MAC does. Zero Collision MAC (ZC-MAC) [28], an earlier effort aimed at enhancing MAC, seeks to achieve zero collisions by breaking down the medium into a predetermined number of channels with size equals to the number of nodes and remembering the slots that clashed in the previous cycles.

The authors in [29] demonstrated a power contention-based ADMC-MAC technique. Since this protocol increases the data transmission speed according to the traffic circumstances and node queue size and enhances energy efficiency, it is simpler to apply to mission-critical applications. The MAC protocol's vulnerability is fixed by this protocol. There are two proposed algorithms in this protocol. A node with many frames in the queues and the most power among the nodes in the virtual cluster is chosen as the cluster head in the first method, which is priority-based. Based on the traffic conditions and remaining energy, the second method uses a predictive technique to predict a node's duty cycle.

The authors of [30] offer a centralized method for scheduling Time Slotted Channel Hopping (TSCH) time slots while making the best use of the available resources. The proposed method is based on the entire sub-graphs derived from the collision matrix of the topology. It is possible to schedule sub-links graphs for the same time period but on different channels at the exact same time. Personal Area Network Coordination (PANC), which is meant to have a thorough understanding of the topology, is used to carry out the suggested technique. A Markov model for predicting the communication time and energy consumption during the frame transmission has also been created using the TSCH MAC. Performance-wise, the proposed technique outperforms earlier equivalent systems.

The study [31] presented a non-conflicting signal scheduling solution based on the association order and outlined the signal slot collision problem. This work also established and distributed multichannel Predetermined and Synchronized Multiple channels Enhanced version Time Slots (DSME-GTSs) scheduling, which efficiently distributes DSME-GTSs over several channels. The goal is to maximize the use of available channels while utilizing the fewest possible time periods. Through the simulations, the effectiveness of the proposed mechanisms is examined in terms of energy saving, transmitting overhead, schedule efficiency, throughput, and latency, and it is established that they exceed existing systems.

In a square region, wireless sensor nodes are dispersed at random. Every simulation involves moving the sink (BS). Placed sensors can sense and gather data without being affected by their geographic position. In the field, sensors have the same energy. Sensors are always transmitting data, and the k-means technique is used to cluster them initially. The distance between the nodes can be used to evaluate the nodes' sleep/awake states. Because some nodes may meet in a probability sampling of the nodes, it can make sense to put a particular node to sleep. Algorithms for machine learning can be used to choose CH more quickly [32].

Agricultural surveillance is accomplished by opportunistically taking use of television white spaces using cognitive radio-based WRAN technology. The spectrum of existing devices can be used without incurring any costs thanks to WRAN technology. CPEs do

not obstruct existing operations because WRAN technology uses base stations that are cognitive radio-based. The proposed network offers wireless communication capabilities for rural agriculture monitoring without incurring any spectrum costs. The proposed network has a bandwidth utilization of 13.4 Mbits and an average energy of 1.02 J. These findings outperform comparable WRAN technology research [33].

The purpose of this research [34] is to investigate a few wireless sensor network MAC-Protocols that are energy-efficient. This research also examines the MAC-Protocols' performance in terms of energy efficiency under various conditions.

Unlicensed users are suggested to employ a novel strategy whereby, when the nodes are in an idle mode, they act as a cooperative relay as explained in [35]. In addition to the suggested method, unlicensed users assist sensor nodes as a cooperative relay while the nodes are in standby mode. Similarly to this, unlicensed users can detect free frequency bands when the nodes are in sleep mode with the aid of the sensor nodes. The suggested cooperative relay makes use of an amplified and forward-based cooperative communication protocol to prevent any disturbance to which remote users may be subjected as a result of the signal attenuation.

Summary of existing protocols is given in Table 1.

**Table 1.** Standard MAC protocol strategies.

| Reference Paper | Approach | Contribution | Results |
|---|---|---|---|
| [4] | Sensor-MAC (S-MAC) | Proposed that nodes in channels used for signaling and message delivery periodically sleep. | Proposed that nodes in channels used for signaling and message delivery periodically sleep. |
| [5] | B-MAC | Low-power listening. | Terrible performance in heavy traffic overheard. |
| [6] | Medium-access control with adaptive sleeping | Based on the absence of activity for a time threshold TA, it changed its duty ratio to adaptive. | Outlined the restrictions of the current S-MAC. |
| [7] | DynMAC | Learned the likelihood of choosing transmission slots depending on the success and collision. | Minimized collisions. |
| [8] | Timeout-MAC (T-MAC) | Allowed for adaptable active/sleep duty cycles depending on hearing for time periods TA and sleeping in the absence of an event. | Minimized collisions and energy consumption. |
| [9] | LWT-MAC | Better energy efficiency and power savings than B-MAC. | Compared to other protocols, lower throughput. |
| [12] | Duty-cycled MAC | Arranged into slots and run according to schedules. The wireless sensor coordinator would be a cluster node and the remaining nodes could be prioritized in accordance with their application requirements. | Improved the delay performance because of the sleep schedules, loose synchronization, and data forwarding sink optimization. |
| [13] | LMAC | Introduced low synchronization overheads and a fault tolerance method. | Did not take resource use into account. |
| [14] | TDMA-based MAC | Organized into slots and run according to schedules. | Queued packets were transmitted in a burst to reduce delays and achieve loose synchronization. |
| [16] | Prioritization-based slotted-CSMA/CA | Organized into slots and run according to schedules. | High throughput, low power consumption, reasonable latency, and flexibility. |
| [17] | Modified CSMA/CA | After a successful transmission, deterministic back-off was enabled. | Decreased the frequency of collisions. |

**Table 1.** *Cont.*

| Reference Paper | Approach | Contribution | Results |
|---|---|---|---|
| [18] | YMAC | Enabled the sleep/wake time scheduling table for nodes, separating the channels into a number of TDMA slots, some of which were reserved for sub-node contention using CSMA/CA. | Increased energy conservation and network throughput. |
| [19] | LEACH-K | Based on TDMA slots the nodes transfer the data to the neighbouring nodes. | No need for global knowledge, an additional overhead for dynamic clustering, or distributed protocol. |
| [23] | Game theoretic MAC protocol | A game theory model was used to adjust the contention window for each node. | The system's throughput rose while its delay and the packet-loss rate decreased, and its energy consumption remained comparatively low. |
| [27] | Hybrid MAC protocol using scheduling-based dynamic sleeping | Enabled the sleep/wake time scheduling table for nodes, separating the channels into a number of TDMA slots, some of which were reserved for sub-node contention using CSMA/CA. | Increased energy conservation and network throughput. |
| [28] | Zero-collision MAC (ZC-MAC) | Zero collisions were based on the medium decomposing to a pre-determined number of slots with the same number of nodes as the slots that interacted during the previous times. | ZC performed better at both high and moderate loads than CSMA and TDMA. |
| [29] | ADMC-MAC | Enhanced energy efficiency and the data transmission performance dependent on the traffic situations by accounting for the size of the node queue. | Enhanced the remaining energy savings as ADMC-MAC. |
| [30] | TSCH MAC | Presented a centralized method to allocate resources efficiently while scheduling TSCH time slots | Superior in performance to earlier similar systems. |
| [31] | DSME-GTSs | A quasi-beacon scheduling method based on the association order was proposed in order to solve the problem of beacon slot collisions. | Maximized the use of available channels while minimizing the number of time periods used. |
| [32] | Cluster head selection algorithm for wireless sensor networks | Placed sensors sensed and gathered data without being affected by their geographic position. In the field, sensors had the same energy. Sensors were always transmitting data, and the K-means technique was used to cluster them initially. | Algorithms for machine learning could be used to choose CHs more quickly. |
| [33] | Sensor network-based opportunistic spectrum utilization for agricultural monitoring | The network offered wireless communication capabilities for rural agriculture monitoring without incurring any spectrum costs. | The network architecture would be implemented into practice in order to monitor agriculture in a real-world rural setting. |

The review's findings revealed a dearth of studies into the cutting-edge techniques suggested to improve the effectiveness of the MAC layer in WSNs. The performance measures and the improvement procedure employed in each strategy are mostly covered in this study.

## 3. Materials and Methods

When compared to other current MAC protocols, the LELLMAC protocol is intended to provide greater control over idle listening, collisions, and the hidden terminal issue, as well as to generate reduced latency. This protocol makes use of three algorithms, namely, the staggered scheduling method, the logical link choice algorithm, and the adaptive reception algorithm, to accomplish its objectives. Detailed explanations of each algorithm are provided in the next section.

### 3.1. Staggered Scheduling Algorithm

In this method, the nodes in the network are made to function in three distinct modes, which are most likely sleep, receive, and transmit, to save energy. After waking up, each node goes to sleep one by one at a predetermined period in order to save energy. When each node wakes up, it listens for communication. It is possible to reduce to a certain degree the amount of energy lost by idle hearing by turning off the radio when the node is asleep. The same schedule is provided to all nodes with the same layer-count as a point of reference, but the sending and receiving time period is collected layer by layer such that while one node is in sending mode, its lower hop adjacent layer node switches to receiving mode. When a message from an upper-layer node (nth layer) is received, each node in the sequential sending period sends the message to the lower hop neighbour layer node. It is possible to reduce end-to-end latency by ensuring that packets are delivered to sink nodes as quickly as possible via multi-layers. As seen in Figure 1, the network's nodes' modes are shown schematically at different levels of the graph.

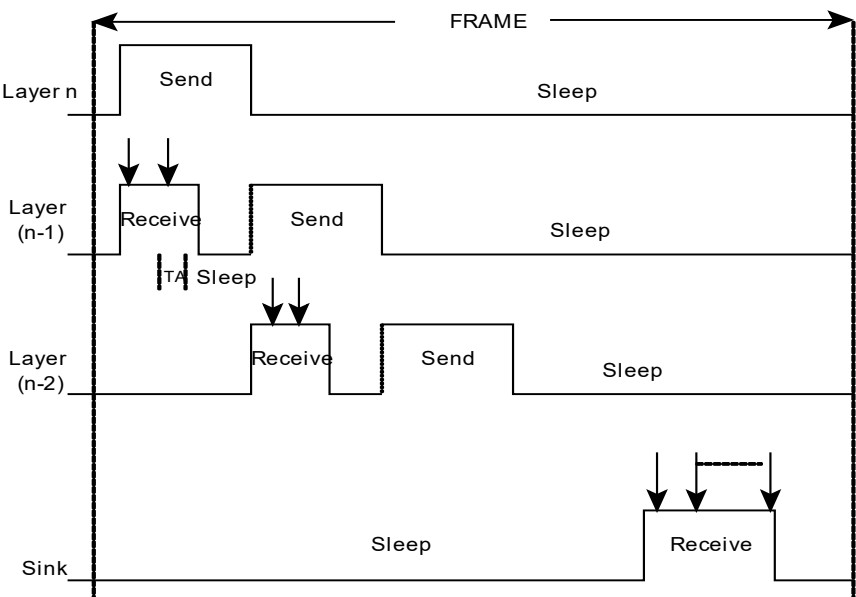

**Figure 1.** Schematic Representation of Staggered Scheduling.

The staggered scheduling algorithm has four advantages:

1. As the nodes in the route rise to their destination one by one, they transmit a packet to the next hop, thereby eliminating the sleep delay.
2. A process with a longer active time may be piped all the way down to the sink, allowing every other node on the multi-hop route to have a higher duty cycle, which will prevent the data from being trapped in the intermediate nodes.;
3. The active periods are now isolated from one another, resulting in less conflict and faster processing times overall.
4. To conserve energy, only the nodes on a multi-hop route are required to raise their duty cycle, while the other nodes may continue to function at their default low-duty cycle.

Suppose if there is an n-layer network design, as shown in Figure 2. The source nodes are found in layer n. In order to clear any possible congestion in the final row of the nodes, many sinks are used. There will be one sink node for every group of three nodes in the final row because of the way the topology is built.

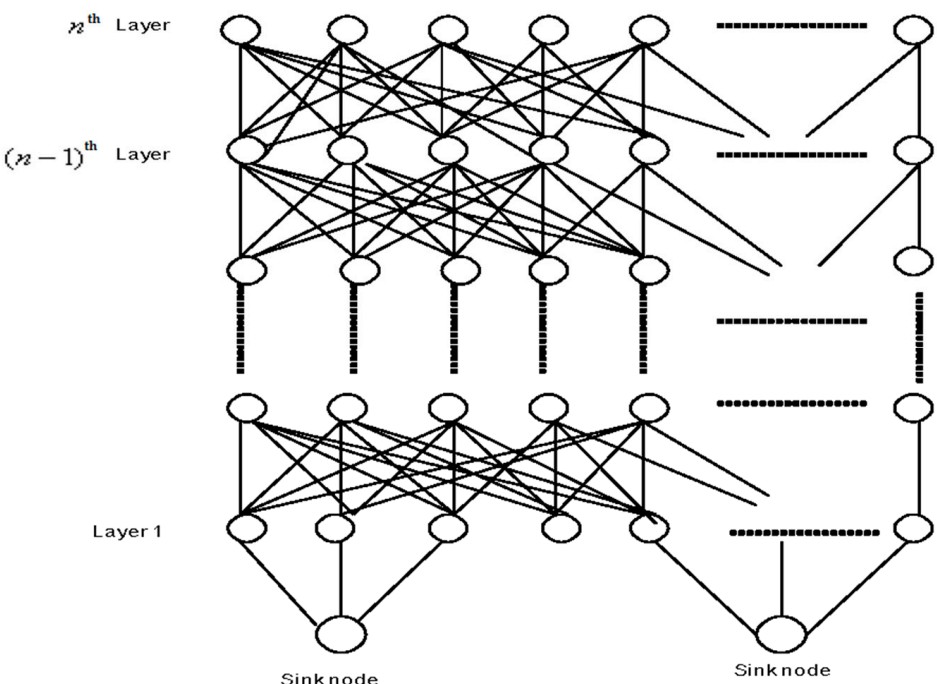

**Figure 2.** Layered Network Structure.

Layer-n is where the source nodes are in the flow diagram of the staggered scheduling method shown in Figure 3; Nodes in the n^th layer are initially used for transmitting mode at a rate of 60 m/s, while those in the layer (n-1) are utilized for receiving mode at a rate of 60 m/s, and so on. Those who stay on the network are in a state of slumber. The source nodes produce packets because of this activity (layer-n). Because the receiving mode is active on the next layer $[(n-1)]$ ^th layer nodes, the source nodes may send packets without first verifying the status of the lower hop neighbour layer nodes, and the transmitted packets are stored in the buffer space. When the reception period of the $[(n-1)]$ ^th layer's nodes come to a close, the same layer begins to transmit data packets to the next low hop neighbour layer nodes in the chain of transmission. As a result, the nodes in the layer are transitioning from sleep to reception mode now. A sleep mode has been activated for the nodes in the residual layer.

After the packets reach the first layer, this kind of staggered scheduling is maintained. Take, for example, the assumption that each node in the first layer has packets to send to the sink: A collision would occur if all the nodes in layer 1 attempted to transmit to the sink at the same time. In this instance, packets are sent to the sink in a round-robin manner, with each node taking turns transmitting to the destination. In order to ensure that the highest priority node transmits first and as quickly as possible, the nodes may be given priority values. A source node does not have to wait for the intended receiver's wake-up before transmitting the packets using this method. Because the receiver node will be listening while the source nodes are in the sending mode, the source nodes may transmit the packets immediately without any hesitation. This would allow for the faster transmission of packets across several levels, thus, decreasing the overall latency from start to finish.

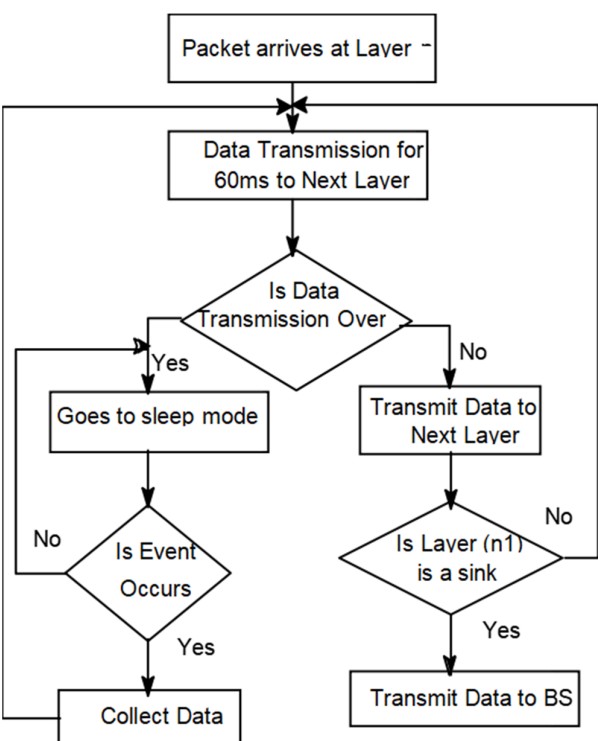

**Figure 3.** Flow Diagram for Stagger Scheduling.

### 3.2. Logical Link Decision Algorithm (LLD)

During the execution of this protocol, the LLD algorithm is used to ensure that two source nodes do not send packets to the same recipient at the exact same moment. In the beginning, when each source node selects a connection to the sink, this method is implemented. After each source node has determined its connections, the links are compared to ensure that there are no overlaps between the relationships already in place. The LLD algorithm determines whether or not there is an overlap and determines whether or not a new connection to the sink is needed. Because no two source nodes may broadcast to the same receiver simultaneously, there will be no collisions between the two sources. By using this method, the network can successfully guarantee that the concealed terminal issue is avoided, as well as that the energy consumption associated with retransmission is minimized. Every edge node in the proposed grid topology has two lower layer nodes in its coverage, whereas every other node in the proposed grid topology has three lower layer nodes in its coverage. The flow diagram for the LLD algorithm is shown in Figure 4. Every source node is utilized to identify a route to the sink by choosing a lower hop neighbor layer node under its coverage at random from the neighboring layer nodes beneath its coverage.

For the messages to travel in the correct direction, it is not necessary to utilize the same connection every time. Each node in a layer receives the final link sequence, as well as links to the previous source nodes in the row, which are then sent to the next node in the layer. Each node checks its connection with the links from the sources that came before it to see if there is any mismatch. If there is any connection overlap, the node will pick a random lower hop layer node from its coverage if there is any link overlap. As a result, this data exchange system is capable of combining collision avoidance and data transfer with high reliability.

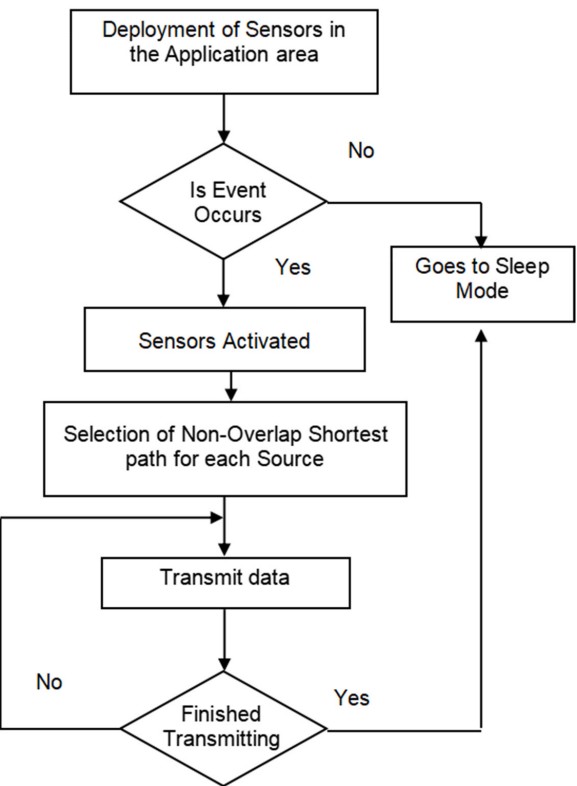

**Figure 4.** Flow Diagram for Logical Link Decision Algorithm.

### 3.3. Adaptive Receiving Algorithm

The flow diagram of the adaptive receiving method is shown in Figure 5. Request to Send (RTS) packets from the $n^{th}$ layer is sent to the $(n-1)^{th}$ layer, while the Clear Clear to send (CTS) signal is sent from the lower hop neighbor layer's node to the higher layer (n). With its response to the CTS signal from the $(n-1)^{th}$ layer, the tenth layer transmits the data to the $(n-1)^{th}$ layer for 60 msec, which is the maximum amount of time allowed. For example, a node in the $(n-1)^{th}$ layer would enter a sleep mode if it detects an idle channel and no data from the layer it is receiving or transmitting data for a period of time while the node is receiving data. Consider the following scenario: If the layer detects any data from the top layer within 60 ms, it will wake up and receive the information. Because of this adaptive reception method, the number of packets dropped is reduced, and the network's throughput is increased. The adaptive receiving system makes use of an interval to deal with varying traffic.

Since the source is situated far from the destination node hence it should have longer than those more drawn out time frame than the hubs situated close to the sink hubs. This is because of the way that a hub that is far away from a sink hub may not hear the RTS that starts a correspondence with its neighbor since it is out of reach; subsequently, the stretch should be sufficiently long to get essentially the start of the CTS bundle. Since the amount of the RTS bundle length and replacement time should be however and is verbalized in condition (1),

$$T_A = R + T \tag{1}$$

Using the above equation, the lowest limit on the length of the interval is determined when it comes to multi-hop packet transmission.

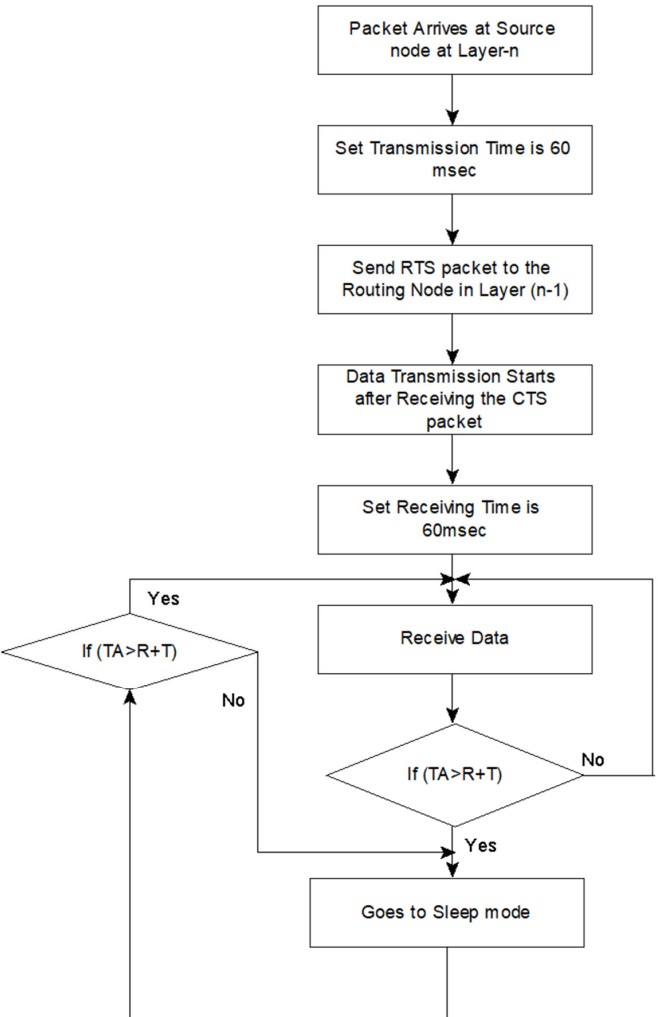

**Figure 5.** Flow Diagram for Adaptive Receiving.

## 4. Results

This segment compares the proposed LELLMAC protocol with the S-MAC protocol in terms of energy consumption, throughput, and latency.

### 4.1. Analytical Model for Energy Consumption

This section provides content on the whole amount of energy used by each node. The real energy consumption, according to [36], is the total of the energy used during the wake-up radio and the energy used during the data transfer.

$$E_{Total} = E_{Wakeup-radio} + E_{data-tx} \qquad (2)$$

(i)   S-MAC Protocol

Whenever a sensor node detects no activity in any event in the environment, the total energy consumption is the same as the total energy ingestion of the wake-up radio (in this case, zero). Assume that a sensor detects a series of events, the packets are to be sent across a hop distance between the sensors. The wake-up signal is delivered at the same time as the sleep signal to wake up the sensor node from sleep mode. This time period is taken into consideration for the computation of energy, which is provided by the equation,

$$E_{wakeup-radio} = (T_{active} \times E_{idle}) + \left( \frac{T_{total}}{2} \times K \times N \times E_{tx} \right) + E_{sync} \qquad (3)$$

where $T_{active}-$ denotes the total amount of time spent in the active state of the wake-up radio for the S-MAC protocol throughout the entire lifespan of the nodes.

The energy used by the radio in the idle, transmitting and receive states is denoted by the variables $E_{idle}$, $E_{tx}$ and $E_{rx}$ respectively. It is the amount of energy needed to achieve periodic synchronization among the nodes that is measured by $E_{sync}$. To compute the total energy consumed by a node the following equation is given by,

$$T_{total} = (T_k P_t + R_k P_r + \beta_t P_l + (1 - (\alpha + \beta)) \times T \times P_S \qquad (4)$$

Here $T_k$ and $R_k$ values represent the number of messages has sent and received in a given an assigned time period. The power used in sending and receiving the packet is denoted by the letters $P_t$ and $P_r$, respectively. $t$ represents the idle time of the node; since the idle time has a constant period, the power consumption during this period will also be constant throughout this period. $(1 - (\alpha + \beta)) \times T$ is the time deployed in sleep mode by a node, while its power used in sleep mode is denoted by $P_s$, and the time required to transmit a data packet is denoted by $T_{data}$. The equation for energy usage during the data transfer may be found here (5)

$$E_{data-tx} = T_{data} \times K \times N \times (E_{tx} + E_{rx}) \qquad (5)$$

(ii)　LELLMAC Protocol

In this protocol, the sensor node communicates with the next lower hop neighbor node without having to constantly transmit the wake-up signal. Consequently, the energy consumption of this protocol is mostly determined by the time it takes to transmit the message signal followed by the acknowledge signal. $T_1$ and $T_2$ refer to the amount of time it takes to transmit the request (RTS and CTS) and ACK packets, respectively.

$$E_{wakeup-radio} = (T_{active} \times E_{idle}) + ((T_1 + T_2) \times K \times N \times E_{tx}) \qquad (6)$$

The amount of energy used all through facts transmission is the same as that of the S-MAC protocol, as shown in Equation (5). The S-MAC protocol and advised LELLMAC protocol power consumption can be anticipated using the Equations (3) to (6), and the analytical representations of those protocols are provided in Figure 6. While the range of nodes is 10 and 50, respectively, it's miles inferred that the energy ingestion of the LELL-MAC protocol is 22% and 76.9% much less than that of the S-MAC protocol with periodic sleep, respectively.

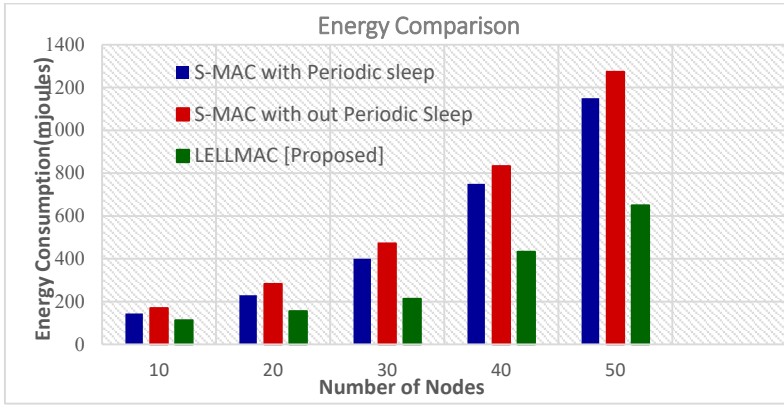

**Figure 6.** Energy Consumption based on Number of Nodes.

### 4.2. Analytical Model for the End-to-End Latency

The end-to-end latency is ordinarily determined by the time taken the packets reached from source to destination. The time it takes to transmit the request and stated packets is represented by way of $T_1$ and $T_2$. Expect that the records sink is $N$ hops distant from

the deliver node $L_s$, and the information packet relay from the source to the sink node is finished through multi-hop communication. For simplicity, assume that the latency on the source node is $L_{source}$, that the latency on the $i^{th}$ intermediate node is $L$, and that the cease-to-quit latency for lots of hops is $L_{in}$, which may be calculated using a mathematical system. Channel set-up time is much like the wake-up radioactive time of the subsequent hop, occasion detection time, and of the subsequent hop, all at the time equal to $T_{sleep}/2$ is the time it takes for every node to accumulate its course inside the communication of the state of the scenario.

(i)   S-MAC Protocol

The latency at the source node $L_s$ may be calculated in (7) while the latency at the intermediate node is the same as the latency at the source node, which is provided by the equation:

$$L_s = \frac{T_{sleep}}{2} + T_1 + T_2 + T_{data}$$

(latency at the reference node = latency at the intermediate node)          (7)

The method for calculating the give latency for a multi-hop transmission is given by means of,

$$L_m = \sum_{i=1}^{N} L_i \qquad (8)$$

(ii)   LELLMAC Protocol

As shown in Equation (7), the latency for the source node Ls is identical to the latency for the S-MAC protocol; however, the set-up time at the in between nodes is removed using the Logical Link Decision (LLD) algorithm. The time length of transmission for the control packets is also not included since such signals are sent consecutively during the time duration of data transmission for the preceding hop, and therefore are not included.

Figure 7 depicts the latency performance for the proposed LELLMAC protocol, S-MAC without sleep, and S-MAC with 10% adaptive sleep for the inter path records transfer. The give up-to-cease latency of the S-MAC with 10% adaptive sleep increases indefinitely as the range of hops will increase. The subsequent hop node should be woken up on the source node, and the same method is accompanied for the intermediate nodes. This method is used in the S-MAC protocol. As a result, the channel st-up time at each of the supply and intermediate nodes is the equal. The propagation delay at each hop includes each of the time it takes to ship the statistics and the time it takes to set up the path. The S-MAC supply node with the no sleep protocol that makes use of uses the same approach as the S-MAC source node with the periodic sleep protocol. There is presently no sleep mechanism applied on the intermediate nodes, which means that all of the node's radios are constantly in the awake kingdom. As a result, the packets of statistics relay from the source to the sink with the least amount of latency whilst compared to the S-MAC with the periodic sleep mechanism. Within the proposed LELLMAC protocol, each supply first chooses the route to the sink for the entire length first. This method ensures that there is no channel setup postpone while the facts is sent across a multi-hop community. The channel setup postpone on the supply node is similar to the S-MAC protocol's delay. As a result, starting with the second hop, the postpone includes the handiest time required for information transmission in place of the time wished for the channel status quo. The provided LELLMAC protocol has an end-to-end latency that is which are 84.7% and 87.4% shorter than the contemporary S-MAC protocol with periodic sleep while the wide variety of nodes is 10 or 50, respectively.

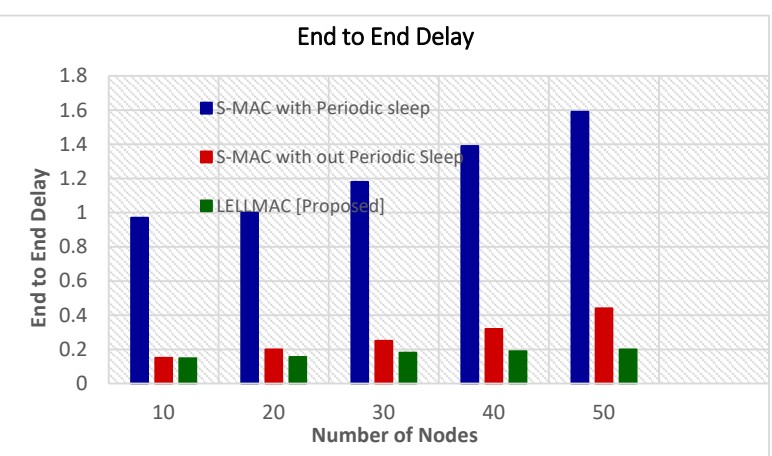

**Figure 7.** End-to-End Latency Comparison Based on Number of Nodes.

*4.3. Analytical Model for the Throughput*

(i)    S-MAC Protocol

In the S-MAC protocol, each sensor node enters sleep mode in one of three circumstances. According to [37], the first instance corresponds to planned sleep time, the second case corresponds to getting an RTS frame from its adjacent nodes, and the third case corresponds to receiving a CTS frame from its surrounding nodes. During the second and third instances, the node will sleep for the duration of a data transmission period that is recorded in the RTS or CTS frames, respectively. As a result, in the S-MAC protocol, a node may only transmit the packets to one other node at a time during a frame period. As a result, the throughput is represented by:

$$Th_{s-MAC} = \frac{n_p}{t_{active} + t_{sleep}} \tag{9}$$

(ii)    LELLMAC Protocol

A node may send packets to the maximum number of nodes during a frame period in the LELLMAC protocol, and therefore the throughput is determined by the formula:

$$Th_{LELLMAC} = \frac{n_p \times n_m}{t_{active} + t_{sleep}} \tag{10}$$

According to Equations (9) and (10) the LELLMAC protocol outperforms the S-MAC protocol in terms of throughput and reliability. Assume that the active time is fixed and equivalent to the number of slots in the LELLMAC protocol in order to determine the successful data transfer there are nodes competing for medium slot resources. A node may only send one request message at a time, and this message is spread evenly over the active time period. The binomial distribution describes the chance that the nodes are located in a particular slot. Due to the fact that the same binomial distribution is applied to the slots, the expected number of slots containing nodes in a slot may be calculated as follows:

$$P[X = n] = \binom{|N}{|n} \left(\frac{1}{S_m}\right)^n \left(1 - \frac{1}{S_m}\right)^{N-n} \tag{11}$$

The same binomial distribution is also applied to the slots thus the excepted number of slots with nodes in a slot is given by,

$$E[X = n] = S_m \left(1 - \frac{1}{S_m}\right) \binom{|N}{|n} \left(\frac{1}{S_m}\right)^{n^{N-n}} \tag{12}$$

Here, $S_m$ represents the number of slots being filled with exactly nodes. So, the average number of collided message is given by,

$$\sigma = \sum_{n=2}^{N} nS_m \binom{|N}{|n} \left(\frac{1}{S_m}\right)^n \left(1 - \frac{1}{S_m}\right)^{N-n} \qquad (13)$$

After the simplification,

$$\sigma = N - N \left(1 - \frac{1}{S_m}\right)^{N-1} \qquad (14)$$

For the LELLMAC protocol, the ratio of the quantity of correctly communicated request messages to the total wide variety of efficaciously transmitted request messages can be computed throughput the usage of Equations (13) and (14) as shown in the table. The throughput performance of the proposed LELLMAC protocol and S-MAC protocol is shown in Figure 8. The successful data transmission ratio $\gamma$ is given by,

$$\gamma = \frac{N - \sigma}{N} = \left(1 - \frac{1}{S_m}\right)^{N-1} \qquad (15)$$

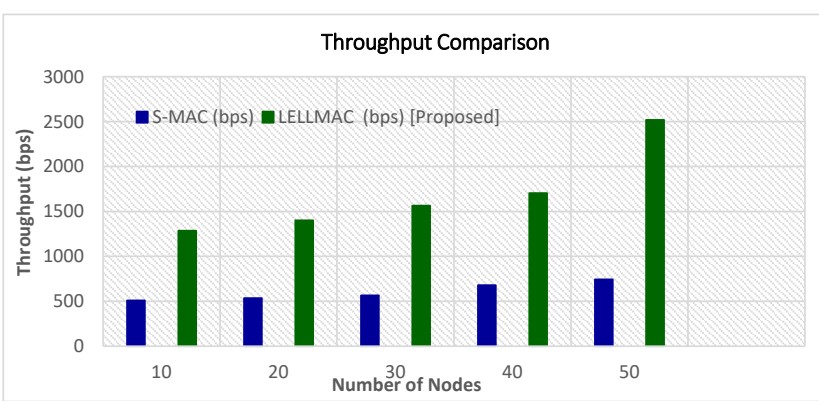

**Figure 8.** Throughput Comparison.

When the number of nodes is 10 or 50, the proposed LELLMAC protocol has a throughput of 60.3 percent and 70.5 percent higher than the current S-MAC protocol respectively.

## 5. Discussion

This section contains the detailed simulation findings and comparisons of the performance of the proposed LELLMAC protocol and other common MAC Protocols, as well as recommendations for further research. The S-MAC protocol was chosen as a baseline for comparison with the proposed LELLMAC protocol since it is generally recognized and popular among the sensor network protocols. Simulations are carried out under a variety of conditions, including the number of nodes, message inter arrival times, and different traffic patterns, as well as the network topology itself (with active and failure nodes). This section is primarily concerned with the energy consumption, throughput, latency, and power efficiency of the system under consideration.

### 5.1. Simulation Set-Up

With the help of the ns2.29 simulator, we were able to build the proposed Lifetime Extension Low Latency MAC protocol (LELLMAC). This comprehensive simulation is used to analyse the performance of the LELLMAC protocol and to compare it to the execution of the S-MAC protocol, both of which are run with the same simulation settings and with a fixed network size in this study. The simulation parameters are provided in Table 2, Table 3 depicts the radio power consumption during the send, receive, and sleep phases of operation.

**Table 2.** Simulation Parameters.

| Parameters | Value |
| --- | --- |
| *Number of nodes* | *100* |
| *Sensing range* | *100X100* Sq.m |
| *Sensing range* | *16* m |
| *Initial energy* | *2* KJ |
| *Sending and receiving slot* | *60* m Sec |
| *Transmission range* | *20* m |
| *Packet size* | *64* Bytes |
| *Energy threshold* $E_{th}$ | *0.001* mjoules |
| *Channel frequency* | *2.4* GHz |
| *Path loss model* | *Two Ray Model* |

**Table 3.** Radio Power Consumption.

| Mode | Power (mW) |
| --- | --- |
| Transmit | 42 |
| Receive | 29 |
| Sleep | 100 μW |
| Idle | 12.36 |

### 5.2. Simulation Tool

A discrete, event simulator called Network Simulator, sometimes known as ns2, is used to represent the wired and wireless network environments. Version 2.29 of the network simulator is utilized in the work that is being presented work. For testing and modelling the behavior of the protocols implemented in C++, the simulator is written in object-oriented C++ and uses OTcl (object Tcl) shell scripts as a front-end tool. Network models are generated by building examples using tcl (tool command language) which are linked to the equivalent C++ object modules. C++ and OTcl have mirrored implementations with a one-to-one relationship.

With the nodes acting as the physical entities in a network and the nodes connected to a collection of protocols as the agent, representing which represented the software entities, Tcl aids the creation of the network typologies. The basic unit of exchanging data among the entities in the simulator is a packet. The simulator offers uni-cast, multicast, and broadcasting packet exchange protocol implementations. All occurrences in the simulators are queued according to time; a scheduler then runs the event that is next to the oldest event inside the queue, finishes the execution of the current event, and adds events periodically as a result of the completed events.

### 5.3. Simulation Scenario

For simulation purposes, 100 nodes are equitably placed in a metre area grid, as shown in Figure 9. This is a five-layer network with ten nodes per layer and a large number of sink nodes in each tier. Each node's coverage range has been set to 16 metres. In this case, each upper layer edge node is covered by two lower layer nodes, and the other upper layer nodes are covered by three lower layer nodes. Using a random number generator, each node chooses one of the lower hops neighbor layer nodes in its covers to send its message. The communications are correct, however they do not take the same path each time they pass along. Overcrowding in of the final row of nodes is managed to avoid by allowing multiple sinks to access the same node at the same time.

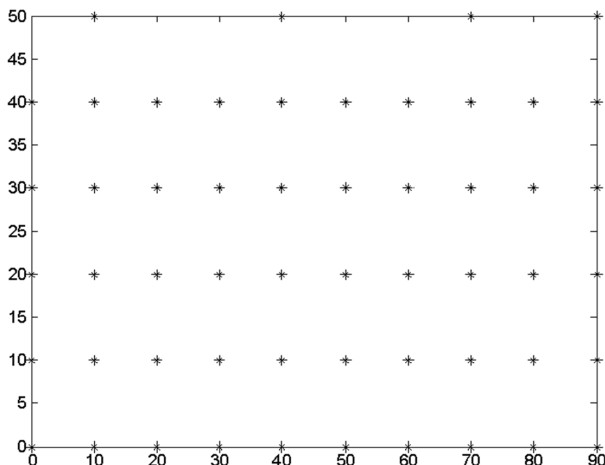

**Figure 9.** Sample Network Topology.

### 5.4. Simulation Results

With the help of the NS-2 simulator, the proposed Lifetime Extension Low Latency MAC protocol (LELLMAC) is developed. In this paper, we investigate the performance of the LELLMAC protocol and compare it to the S-MAC protocol, both of which are implemented with the same simulation settings and a constant network size.

### 5.5. Energy Consumption

Figure 10 shows the change in energy usage as a function of the time between the message arrivals. For synchronization, the transmitting, and receiving periods of each node are both set to 60 ms. The amount of energy used by the network at different message inter-arrival intervals is calculated and displayed in this paper. The energy consumption of a radio in a certain mode may be estimated by multiplying the duration by the amount of power needed to run the radio in that mode.

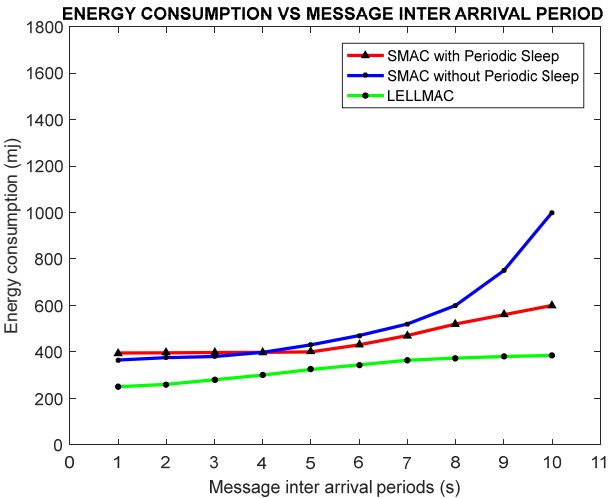

**Figure 10.** Comparison of Energy Consumption Based on Massage Interval Arrival Period.

The plot infers that the energy ingestion of the S-MAC and LELLMAC protocol is shown in the plot. For the sake of this comparison, a simulation configuration with five levels, 10 nodes in each layer, and a four-sink is used. In this graph, it can be shown that the LELLMAC has a lower energy usage than the S-MAC. This is since the idle listening accounts for most of the energy consumption in the S-MAC protocol, although the TA may induce the LELLMAC to enter sleep mode sooner, resulting in a reduction in energy consumption. It uses 55 to 63 percent less energy than the current S-MAC protocol with a periodic sleep protocol, according to the LELLMAC protocol's specifications.

In Figure 11, the energy consumption of nodes with regard to the wide variety of nodes is shown. The energy consumption of the S-MAC with the periodic sleep protocol is higher than that of the LELLMAC protocol and lower than that of the S-MAC without the sleep protocol because, inside the S-MAC protocol, the sensor nodes have to wake up periodically even when there may be no traffic inside the sensor community, ensuing in a better consumption of electricity. Whilst in comparison to the S-MAC protocols, the LELLMAC protocol ingest much less electricity due to the staggered scheduling that is used. While the number of nodes is 10 and 100, the proposed LELLMAC protocol consumes 22 % and 76% less energy consumption than the modern S-MAC protocol with 10 percentage adaptive sleep, ensuing in 22 percent and 76 percent power savings, respectively.

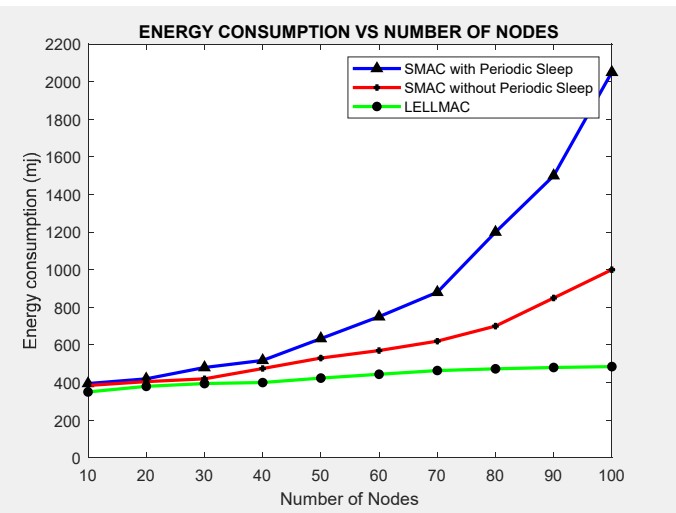

**Figure 11.** Comparison of Energy Consumption Based on Number of Nodes.

### 5.6. Throughput

The throughput comparison of the LELLMAC protocol with the S-MAC protocol is shown in Figure 12. When the traffic load is high, the LELLMAC produces a much greater throughput than the S-MAC. Because there is no adaptive listening in the S-MAC, and because the duty cycle is set in this protocol, the network performance may decrease when there is significant traffic. When the number of nodes is 10 or 100, the proposed LELLMAC protocol achieves an average throughput of 59 percent and 69.3 percent higher than the current S-MAC protocol with periodic sleep, respectively.

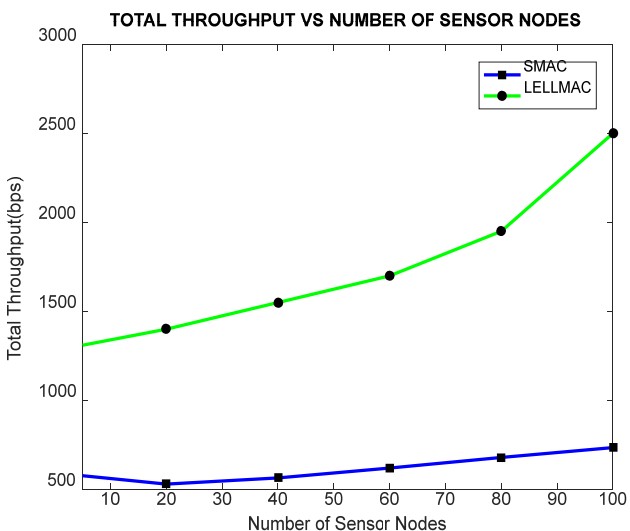

**Figure 12.** Throughput Comparison with SMAC.

### 5.7. Latency

The latency experienced in the LELLMAC protocol as compared to that seen in the S-MAC regarding the number of nodes is shown in Figure 13. Because of the staggered scheduling method, the latency of the LELLMAC is much lower than that of the S-MAC, as is evident. It is necessary for a node in the S-MAC to wait till its neighbor is awake before transmitting a message to it. In contrast, the LELLMAC protocol does not have any delay at all, which is due to synchronization. When a node is in the transmit state in the LELLMAC, the lower layer node is in the receiving state, and vice versa.

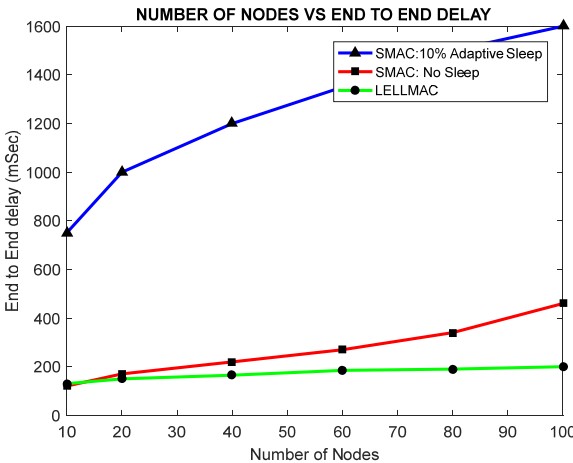

**Figure 13.** Latency Comparison with SMAC.

Consequently, the source node may send the message to the lower layer without first verifying if the lower layer node is listening. As a result, the source node may send its message quickly to the sink by using several levels of communication. Figure 13 infers that when the number of nodes is 10 or 100, the proposed LELLMAC protocol has an end-to-end latency that is 82.3 percent and 86.6 percent shorter than the current S-MAC protocol with 10% adaptive sleep, respectively.

### 5.8. Power Efficiency

Figure 14 illustrates the comparison of power efficiency between the LELLMAC and SMAC protocols. The power efficiency, which is the amount of throughput accomplished per unit of energy used, is calculated as follows:

$$\text{Power Efficiency} = \frac{Total\ \ Throughput}{Total\ \ Energy\ \ Consumption} \tag{16}$$

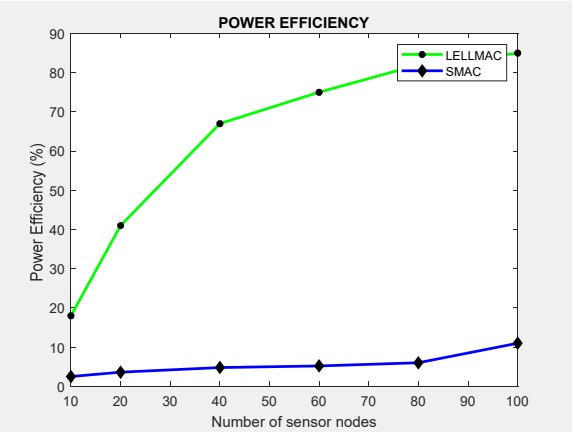

**Figure 14.** Power Efficiency Comparison with SMAC.

Because of the hybrid algorithm employed in the LELLMAC protocol, the suggested protocol has a power efficiency that is 67% greater than that of the S-MAC protocol.

This section may be divided by subheadings. It should provide a concise and precise description of the experimental results, their interpretation, as well as the experimental conclusions that can be drawn.

## 6. Conclusions

Based on the hybrid algorithm for the channel access mechanism, which may substantially decrease the power consumption during communications at the data link layer, we present our findings in this article. Unlike current protocols, an LLD algorithm is intended to choose the best route for data transmission between the nodes from the source to the destination. When compared to existing protocols, an LLD algorithm may decrease the end-to-end latency and packet loss rate among nodes in a regularly deployed node. Analytical models are created to investigate the performance parameters of WSNs, such as latency, data transfer, throughput, and energy usage, among other things. The results of the simulation show that our approach has the potential to reduce energy consumption while simultaneously increasing throughput and decreasing latency. Our suggested protocols may thus be used in a variety of WSN applications, including industrial, commercial, and healthcare environments where energy efficiency, latency, and packet loss rate are important considerations.

**Author Contributions:** Conceptualization, T.H., J.Z. and T.V.P.; Methodology, T.V.P., D.N.A.J., M.A. and A.Q.M.; Formal analysis, D.N.A.J. and M.A.; Investigation, T.H. and A.Q.M.; Resources and data collection, J.Z. and T.V.P. Writing, T.V.P., J.Z. and T.H.; Validation, D.N.A.J.; Funding Acquisition, A.Q.M., T.V.P. and J.Z. All authors have read and agreed to the published version of the manuscript.

**Funding:** This research received no external funding.

**Institutional Review Board Statement:** Not applicable.

**Informed Consent Statement:** Not applicable.

**Data Availability Statement:** Not applicable.

**Acknowledgments:** This work was supported by the National Natural Science Foundation of China (No. 61862051), the Science and Technology Foundation of Guizhou Province (No. [2019]1299, No. ZK[2022]449), the Top-notch Talent Program of Guizhou province (No. KY[2018]080), the Natural Science Foundation of Education of Guizhou province (No. [2019]203) and the Funds of Qiannan Normal University for Nationalities (No. qnsy2019rc09).

**Conflicts of Interest:** The authors declare no conflict of interest.

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
