# Peer review of "Design and Validation of Lifetime Extension Low Latency MAC Protocol (LELLMAC) for Wireless Sensor Networks Using a Hybrid Algorithm"

_sustainability, doi:10.3390/su142315547_

Round 1

Reviewer 1 Report

The paper proposed a hybrid algorithm to improve the energy efficiency of sensor networks with nodes that are regularly placed.

The paper is well written.

I have some recommendations.

Some numerical values should be given at the end of abstract.

Related works should be presented in a separate section.

Simulation software and newtork environment should be described.

More simulation parameters should be added such as bandwidth, data rate, etc.

Some legends in figures are difficult to read.

Some paper should be referenced as below.

https://doi.org/10.1080/00207217.2019.1636313

https://doi.org/10.1016/j.suscom.2020.100404

https://doi.org/10.1007/s11277-021-08990-3

Author Response

Some numerical values should be given at the end of abstract.

Response: Thank you very much for your valuable suggestions; we have incorporated the changes in the revised manuscript and we have included numerical values in the abstract.

Related works should be presented in a separate section.

Response: Thank you very much for your valuable suggestions; we have incorporated the changes in the revised manuscript and the introduction and related works are addressed separately. Introduction is presented in section 1 and related works is presented in Section 2.

Simulation software and newtork environment should be described.

Response: Thank you very much for your valuable suggestions; we have incorporated the changes in the revised manuscript. Simulation Software is described in Section 5.2 and Network environment is discussed in 5.3

More simulation parameters should be added such as bandwidth, data rate, etc.

Response: Thank you very much for your valuable suggestions; we have incorporated the changes in the revised manuscript. Explained the parameters in section 5.5 and 5.6.

Some paper should be referenced as below.

https://doi.org/10.1080/00207217.2019.1636313

Response: Response: Thank you very much for your valuable suggestions; we have incorporated the changes in the revised manuscript. Included in section 2 and in Reference [35].

https://doi.org/10.1016/j.suscom.2020.100404

Response: Response: Thank you very much for your valuable suggestions; we have incorporated the changes in the revised manuscript. Included in section 2 and in Reference [33].

https://doi.org/10.1007/s11277-021-08990-3

Response: Response: Thank you very much for your valuable suggestions; we have incorporated the changes in the revised manuscript. Included in section 2 and in Reference [32]

Reviewer 2 Report

Abstract

Abstract should be review clarifying the existing problem to be solve and the approach of the LELLMAC. Avoid to used short and generic sentences like “The MAC protocol design solves this issue”.

Introduction

This section should be organized trying to identify current MAC protocols for energy efficient systems like WSN. At least a clear indication on IEEE 802.15, especially 802.15.4 (ZigBee) should be describe as well as current solutions on 5G like Non-Orthogonal Multiple Access. Links to modern survey analysis like Balobaid or similar papers should be considered.

A.     Balobaid, "A survey and comparative study on different energy efficient MAC-protocols for Wireless Sensor Networks," 2016 International Conference on Internet of Things and Applications (IOTA), 2016, pp. 321-326, doi: 10.1109/IOTA.2016.7562745.

Section should finish with a table summarizing and comparing the different current alternatives.

Section 2. Provide a detail description of LELLMAC. An introduction to LELLMAC protocol should be provided. This is mainly a wording problem trying to organized and provide a friendly description of characteristics, and algorithms.

Graphics Quality is low. Specially Figure 1, 2, 3, 4, 5 and 11. Typo at Figure 7 (Enf à End). Figure 7 caption overlap with bars.

Review in detail all the text as there are typos like: seventy six.9% à 76.9% (line 273). In the text appear acronyms which are not expand and you should not consider that readers of energies are communications experts like CTS (Clear to Send) RTS (Ready to Send),…. Homogenized terminology ej: mj and mjoules

Clarify text on simulator as you claim about ns2.29 simulator and NS-2.

Figure 13 is missing in the text.

New protocol seems to provide a better performance in the three parameters evaluated but paper should also provide disadvantages of the proposed protocol including synchronization and implementation. Please consider an analysis of this at conclusions

References to the state of the art are really old with more that 50% before 2010. This should be revised and update to modern state of the art solutions specially on IoT environment where current proposal is relevant. In addition, a review of the text in relation with reference relevant work should be done including  simulator environment

Author Response

Abstract 

Abstract should be review clarifying the existing problem to be solve and the approach of the LELLMAC. Avoid to used short and generic sentences like “The MAC protocol design solves this issue”.

Response: Response: Thank you very much for your valuable suggestions; we have incorporated the changes in the revised manuscript. Explained in detail in Abstract

Introduction

This section should be organized trying to identify current MAC protocols for energy efficient systems like WSN. At least a clear indication on IEEE 802.15, especially 802.15.4 (ZigBee) should be describe as well as current solutions on 5G like Non-Orthogonal Multiple Access. Links to modern survey analysis like Balobaid or similar papers should be considered.

  1. Balobaid, "A survey and comparative study on different energy efficient MAC-protocols for Wireless Sensor Networks," 2016 International Conference on Internet of Things and Applications (IOTA), 2016, pp. 321-326, doi: 10.1109/IOTA.2016.7562745.

Response: Response: Thank you very much for your valuable suggestions; we have incorporated the changes in the revised manuscript. Included in Section 2 section 2 and in Reference [34]

Section should finish with a table summarizing and comparing the different current alternatives.

Response: Response: Thank you very much for your valuable suggestions; we have incorporated the changes in the revised manuscript. Table 1 Shows the comparison of different existing MAC Protocols.

 Section 2. Provide a detail description of LELLMAC. An introduction to LELLMAC protocol should be provided. This is mainly a wording problem trying to organized and provide a friendly description of characteristics, and algorithms.

Response: Response: Thank you very much for your valuable suggestions; we have incorporated the changes in the revised manuscript. Introduction to LELLMAC protocol is given in Section 3

Graphics Quality is low. Specially Figure 1, 2, 3, 4, 5 and 11. Typo at Figure 7 (Enf à End). Figure 7 caption overlap with bars.

Response: Response: Thank you very much for your valuable suggestions; we have incorporated the changes in the revised manuscript. Changed all the Figure which is mentioned above and Figure 7 caption is also rewritten

Review in detail all the text as there are typos like: seventy six.9% à 76.9% (line 273). In the text appear acronyms which are not expand and you should not consider that readers of energies are communications experts like CTS (Clear to Send) RTS (Ready to Send),…. Homogenized terminology ej: mj and mjoules.

Response: Response: Thank you very much for your valuable suggestions; we have incorporated the changes in the revised manuscript. Corrected all the terms (in Section 3.3) which is given in reviewer comments.

Clarify text on simulator as you claim about ns2.29 simulator and NS-2.

Response: Explained in Section 5.2.

 Figure 13 is missing in the text.

Response: Response: Thank you very much for your valuable suggestions; we have incorporated the changes in the revised manuscript. Figure 13 is included in Section in 5.7.

 New protocol seems to provide a better performance in the three parameters evaluated but paper should also provide disadvantages of the proposed protocol including synchronization and implementation. Please consider an analysis of this at conclusions

References to the state of the art are really old with more that 50% before 2010. This should be revised and update to modern state of the art solutions specially on IoT environment where current proposal is relevant. In addition, a review of the text in relation with reference relevant work should be done including simulator environment.

Response: Response: Thank you very much for your valuable suggestions; we have incorporated the changes in the revised manuscript. In section 2 recent papers reviews are included.

Round 2

Reviewer 2 Report

Authors has improved paper according to remarks. Thanks